# Urinary Metabolomic Differentiation of Infants Fed on Human Breastmilk and Formulated Milk

**DOI:** 10.3390/metabo14020128

**Published:** 2024-02-16

**Authors:** Ji-Woo Yu, Min-Ho Song, Ji-Ho Lee, Jun-Hwan Song, Won-Ho Hahn, Young-Soo Keum, Nam Mi Kang

**Affiliations:** 1Department of Crop Science, Konkuk University, 120 Neungdong-ro, Gwangjin-gu, Seoul 05029, Republic of Korea; wooody96@konkuk.ac.kr (J.-W.Y.); hlhkkl@konkuk.ac.kr (M.-H.S.); micai1@naver.com (J.-H.L.); rational@konkuk.ac.kr (Y.-S.K.); 2Department of Pediatrics, Soonchunhyang University, 30, Suncheonhyang 6-gil, Dongnam-gu, Cheonan-si 31151, Republic of Korea; joonanii@schmc.ac.kr (J.-H.S.); hahn@schmc.ac.kr (W.-H.H.); 3Department of Nursing, Research Institute for Biomedical & Health Science, Konkuk University, Chungju-si 27478, Republic of Korea

**Keywords:** human milk, formula milk, urine, metabolomics, hydroxyhuppiric acid

## Abstract

Human breastmilk is an invaluable nutritional and pharmacological resource with a highly diverse metabolite profile, which can directly affect the metabolism of infants. Application of metabolomics can discriminate the complex relationship between such nutrients and infant health. As the most common biological fluid in metabolomic study, infant urinary metabolomics may provide the physiological impacts of different nutritional resources, namely human breastmilk and formulated milk. In this study, we aimed to identify possible differences in the urine metabolome of 30 infants (1–14 days after birth) fed with breast milk (*n* = 15) or formulated milk (*n* = 15). From metabolomic analysis with gas chromatography-mass spectrometry, 163 metabolites from single mass spectrometry (GC-MS), and 383 metabolites from tandem mass spectrometry (GC-MS/MS) were confirmed in urinary samples. Various multivariate statistical analysis were performed to discriminate the differences originating from physiological/nutritional variables, including human breastmilk/formulate milk feeding, sex, and duration of feeding. Both unsupervised and supervised discriminant analyses indicated that feeding resources (human breastmilk/formulated milk) gave marginal but significant differences in urinary metabolomes, while other factors (sex, duration of feeding) did not show notable discrimination between groups. According to the biomarker analyses, several organic acid and amino acids showed statistically significant differences between different feeding resources, such as 2-hydroxyhippurate.

## 1. Introduction

Breastfeeding is integral to the healthy growth of newborns, delivering complex bioactive components like glycerolipids, antibodies, and antioxidants [1,2,3]. However, challenges arise when mothers face chronic diseases or produce insufficient breast milk, necessitating the use of formula milk [4,5]. In this context, it becomes crucial for formula milk not only to meet nutritional needs but also to mirror the bioactive compounds found in breast milk. Ongoing research is dedicated to enhancing the nutritional composition of infant formula, aiming to replicate the numerous health benefits associated with breastfeeding [6,7]. These studies focus on optimizing formula content for optimal growth and development, striving to provide formula-fed infants with comparable health outcomes, including a bolstered immune system and enhanced cognitive development.

Analyzing human urine can provide valuable insights into an individual’s health status, as it contains a wealth of biochemical information, including metabolites, proteins, and other biomarkers [8]. This non-invasive and cost-effective method of health assessment offers the advantage of early disease detection, monitoring chronic conditions, and assessing the impacts of dietary and lifestyle choices on overall well-being [9]. Research into urine analysis as a diagnostic tool continues to advance, opening new possibilities for personalized medicine and proactive healthcare interventions [10]. Metabolomics, a branch of science that focuses on the comprehensive analysis of metabolites in biological samples, is a powerful method used in urine analysis to unravel the complex metabolic processes occurring in the human body. By employing metabolomics, researchers can uncover patterns and alterations in metabolite profiles that may serve as early indicators of various health conditions, enabling a deeper understanding of an individual’s physiological state through the analysis of their urine [11,12,13].

In this paper, urine metabolomics analysis was employed to investigate the differences between breastfed and formula-fed infants by analyzing their urinary metabolites. The possible impact of breastfeeding was analyzed via urinary metabolism analysis. This study could provide fundamental data to understand the urinary metabolome in neonates related to feeding type.

Additionally, targeted metabolomics was utilized to confirm distinct metabolic patterns between the two groups, including breastmilk/formulate milk feeding, sex, and the duration of feeding. To investigate key biomarkers between the human-milk-fed group (HM) and formulated-milk-fed group (FM), supervised statistical analysis and pathway studies were conducted.

## 2. Materials and Methods

### 2.1. Sample Collection

Urine samples were obtained from neonates, with consent from willing mothers at Soonchunhyang University. This study compared 15 formula-fed and 15 breastfed infants, analyzing urine samples collected from day 1 up to 17 days post-birth. Gender, twin status, and birth timing were distributed similarly in each experimental group to ensure homogeneity. Maternal participants and infants involved in the experiment underwent BMI measurements and blood tests to confirm the absence of specific diseases, and only healthy mothers and infants participated in this study. All the mothers who donated urine for this study did so voluntarily and with consent, and this study was approved by the Research Ethics Committee (Institutional Review Board of Soonchunhyang University Hospital, protocol code SCHCA 2020-08-034, 1 September 2020). Detailed information about urine samples and each neonate’s health are described in Appendix A.

### 2.2. Sample Extraction for Instrumental Analysis

Urine samples of 100 μL were taken and reacted with a urase solution of 10 μL (100 units/10 μL) at 37 °C for 60 min. We added 890 μL of extraction solvent (methanol/formic acid = 99.875:0.125, *v*/*v*) and vortexed it for 5 min. To precipitate extra protein, we centrifuged the sample at 16,100× *g* for 10 min at 4 °C. We took 400 μL of supernatant and dried it in a vacuum concentrator.

Before instrumental analysis, all samples were derivatized with methoxyamine and sylation. In brief, the dried residue was then treated with 50 μL of a methoxyamine reagent (20 mg/mL methoxyamine hydrochloride in pyridine) at 37 °C for 90 min, followed by the addition of 50 μL of N-Methyl-N-(trimethylsilyl) trifluoroacetamide (MSTFA) reagent (MSTFA + 1% TMCS) and a subsequent reaction at 37 °C for 30 min.

### 2.3. Instrumental Analysis

Untargeted metabolomics employed a GCMS-QP2010SE system (Kyoto, Japan) equipped with a fused silica Rxi-5 ms column (30 m, 0.5 μm film thickness, 0.25 mm ID; Restek Corporation, Bellefonte, PA, USA). Helium was utilized as the carrier gas at a liner flow of 36.7 cm/min (5.8 mL/min total flow). The injector and MS ion source were maintained at 260 °C, while the MS interface was configured to 280 °C. The column oven temperature was initially at 150 °C for 1 min, and then it was increased linearly by 25 °C/min until reaching 300 °C, where it was held for 30 min. Samples and standards (1 µL) were introduced at a 1:2 split ratio.

For targeted metabolomics, analysis was conducted using a Shimadzu GC-MS-TQ8040 (Kyoto, Japan) instrument utilizing multiple reaction monitoring (MRM) ions, encompassing a total of 395 metabolites. A 1.0 μL sample aliquot was injected into a BPX-5 column (30 m × 0.25 mm i.d.; 0.25 μm film thickness) in split (40:1) mode. Optimal GC-MS/MS parameters were configured, including an injector temperature of 250 °C and ion source and transfer line temperatures of 200 °C and 280 °C, respectively. The initial oven temperature was set at 60 °C (2 min), followed by a rise to 320 °C (10 °C/min) and a 15-minute stabilization. Helium was employed as the carrier gas at a flow rate of 1 mL/min, and argon was used as the collision gas. The electron ionization energy was adjusted to 70 eV.

### 2.4. Statistical Analysis

MetaboAnalyst 5.0, an online platform for metabolomic analysis, was utilized for conducting multivariate data analyses. Unsupervised principal component analysis (PCA) was performed to evaluate clustering separation patterns among different test groups, while supervised partial least squares-discriminant analysis (PLS-DA) was employed to differentiate between treatment groups. Metabolites with VIP scores > 1 and standard errors < 1 were identified through the PLS-DA’s variable importance in the projection (VIP) scores. Biomarker metabolites significantly affected by exposure to azole pesticides in the high-concentration group compared to the control group were determined via one-way ANOVA, with *p* < 0.05 set as the significance threshold. A heatmap depicting relative areas of chosen biomarker metabolites, based on VIP, ANOVA, and fold changes, was generated using MetaboAnalyst 5.0 (www.metaboanalyst.ca, accessed on 7 February 2023). Metabolic pathway analysis plots were also created using MetaboAnalyst 5.0, and the metabolic pathways were identified through the Kyoto Encyclopedia of Genes and Genomes (KEGG) library.

## 3. Results

### 3.1. Untargeted Metabolomics Using GC-MS

Untargeted metabolomics is a methodology used to comprehensively analyze all metabolites present in a sample without prior target selection [14]. Using this approach, specific metabolites are not predefined, allowing for the discovery of novel metabolites and unique patterns of metabolites that may not naturally occur in the human body or exhibit distinct tendencies in the sample being studied [15].

For the present study, 6 samples from the formulated group and 11 samples from the breast milk group were selected for untargeted metabolomics analysis out of a total of 61 urine samples. Gas chromatography-mass spectrometry (GC-MS) was employed in full scan mode to perform the analysis. The GC-MS analysis identified a maximum of 197 peaks, representing various metabolites such as amino acids, sugars, and fatty acids.

The chromatograms of the human-milk-fed group and formula-milk-fed group exhibited distinct difference patterns. For example, increases in some metabolites such as ascorbate, threonine, and phenylacetate were found in the human-milk-fed group (Figure 1).

To further explore the metabolomic profile, a total of 163 peaks were chosen for statistical analysis (Appendix A). This selection excluded metabolite peaks that posed challenges in quantification or qualitative identification. To conduct statistical analysis, MS total useful signal (MSTUS) approaches were employed for the normalization of urine concentration. This deliberate exclusion of xenobiotics and artifacts ensures a reliable measure of urine concentration [15].

In the statistical analysis, the t-test analysis did not identify metabolites that exhibited statistically significant differences. This observation suggested that due to substantial inter-individual variations and influences from genetic and environmental factors, it is challenging to establish statistical significance in urine samples.

Given the complexity of metabolomics, multivariate statistical analysis methods such as principal component analysis (PCA) and partial least squares discriminant analysis (PLS-DA) are commonly employed [16,17]. In this study, PCA and PLS-DA were utilized to investigate the differences in metabolite patterns between the breast milk and formula milk groups.

While the PCA plot did not display a clear separation between the two groups, there was a tendency of partial separation along PC1 and PC2, accounting for 30.2% and 10.9% of the variance, respectively (Figure 2a,b).

Interestingly, the PLS-DA plot demonstrated a distinct separation between the two groups. PC1 and PC2 explained 26.7% and 12.3% of the variance, respectively, revealing a statistically significant differentiation in urine metabolites between the breast milk and formula milk groups. Among the 163 identified peaks, 60 peaks had VIP (Variable Importance in Projection) values exceeding 1, indicating their substantial contribution to the observed separation (Figure 2c).

However, because the specific identification of these individual metabolites was not achieved, potential biomarkers could not be selected via untargeted metabolomics. Consequently, targeted metabolomics was conducted for a more detailed analysis and exploration of potential biomarkers.

### 3.2. Targeted Metabolomics and Pathway Analysis

Targeted metabolomics is a quantitative analytical approach that aims to measure and quantify a predefined set of metabolites within a biological sample, providing insights into specific metabolic pathways and molecular interactions [18,19]. The utilization of targeted metabolomics in human urine analysis offers the advantage of the precise and focused quantification of predetermined metabolites, enabling a comprehensive understanding of specific biochemical pathways and potential biomarkers [20].

In this study, targeted metabolomics was conducted using GC-MS/MS in MRM mode to simultaneously analyze a total of 395 metabolites. A total of 383 metabolites were identified in urine samples and utilized for multivariate statistical analysis (Appendix A).

Fold-change analysis was conducted to investigate overall trends in metabolites. Contrary to the results of untargeted metabolomics, it was observed that the majority of metabolites (225 metabolites) decreased in the breast milk feeding group, while 27 metabolites increased. In this study, we observed distinct patterns differentiating the breast milk and formula groups through untargeted metabolomics analysis. To identify the metabolites that significantly contributed to this differentiation, we employed targeted metabolomics to pinpoint biomarkers. The results of targeted metabolomics were then emphasized in a statistical analysis using orthogonal partial least square discriminant analysis (OPLS-DA) plots (Figure 3d), which clearly demonstrated a pronounced separation between the two groups.

In summary, untargeted metabolomics is useful for quickly comparing patterns and identifying differences between two groups without specific peak identification. Once differences are established, targeted metabolomics can be utilized to identify biomarkers that influence the separation between the groups.

The detailed investigations were obtained via multivariate analysis. The PCA analysis created a model explaining 26% and 7% of the variance by PC1 and PC2, respectively, though a clear separation between the breast milk and formula milk groups was not observed (Figure 3a,b).

For PLS-DA analysis, a model explaining 25.7% and 3.3% of the variance by PC1 and PC2, respectively, was established, confirming the separation between the breast milk and formula milk groups. The outcomes of OPLS-DA demonstrated a more distinct separation between the two groups, with a Q2 value of 0.38, indicating a reasonably reliable model. This separation implied variations in urinary metabolites between the breast milk and formula feeding groups, suggesting that the type of feeding may influence the metabolism of neonates (Figure 3c,d).

Through the multivariate analyses, important biomarkers were selected, with 167 metabolites with VIP values over one identified as biomarkers. These biomarkers included organic acids, sugars, and amino acids, indicating their importance in distinguishing significant differences between neonates in the breast milk and formula milk groups.

The heatmap of 25 high VIP metabolites in Figure 2 reveals that limited numbers of metabolites such as 2-hydroxyhippuric acid and 2-phosphoglyceric acid were increased in the breastfeeding group, while the levels of 21 other metabolites were down-regulated in the same group. In particular, the t-test showed that 2-hydorxyhippuric acid was significantly increased in the human-milk-fed group (Figure 4).

Utilizing these biomarkers, pathway analysis revealed that alanine, aspartate, and glutathione metabolism, as well as glyoxylate and dicarboxylate metabolism, prominently influenced metabolic processes (Figure 5).

## 4. Discussion

In this study, untargeted metabolomics was employed to reveal comprehensive insights into the urine metabolome, paving the way for potential biomarker discovery, or gain a deeper understanding of metabolic alterations related to specific conditions or diseases. The chromatographic trends and patterns between the breastfed and formula-fed groups exhibited marked differences. These findings suggest distinctions in urine metabolism between infants receiving breast milk and those receiving formula.

Biomarker research was conducted using untargeted metabolomics approaches. In particular, 2-hydroxyhippuric acid was significantly increased in the human-milk-fed group. Hydroxyhippuric acid is a derivative of hippuric acid, which is a highly abundant urinary metabolite associated with the metabolic end product of most polyphenols [21,22]. Urinary hippuric acid and hydroxyhippuric acid are promising candidate biomarkers for liver function, autism spectrum disorder diagnosis, and aromatic compound exposure [23,24,25]. Notably, previous studies have demonstrated a direct positive correlation between urinary hippuric acid and the consumption of flavonoids, particularly in fruits and fruit-derived beverages [26,27]. Breast milk contains various polyphenols, including flavonoids, and the polyphenol content varies depending on the mother’s diet [28]. In this study, the observed elevation of hydroxyhippuric acid, a polyphenol metabolite, in the urine of infants in the breastfed group implied a potential higher intake of polyphenols from human milk compared to formula-fed infants.

In pathway analysis, amino acid metabolisms were selected as impacted pathways such as alanine, aspartate and glutamate metabolism, glycine, serine and threonine metabolism, and phenylalanine metabolism. The amino acids related with these pathways such as serine, glycine and alanine showed decreasing trends in the human-milk-fed group compared to formula-milk-fed group. The secreted amino acid in urine was related to the efficiency of absorbed dietary protein [29]. The efficiency of protein absorption is typically assessed via complex calculations involving protein intake and nitrogen metabolites [29]. However, assuming equal breast milk and formula intake in infants, the observed decrease in amino acid levels in the urine suggests higher protein absorption efficiency in the breastfed group. For more precise validation, future research could leverage protein composition and intake analysis data from breast milk and formula, providing valuable insights for further investigation.

This study had limitations in its sample size, as specimens from only 30 newborns were utilized, making the generalization of the results challenging. Despite this limitation, statistically significant differences were observed in the urinary metabolites of infants who were breastfed compared to those who were formula-fed, with several biomarkers, including hippuric acid, identified as contributors to these differences. This study suggests the potential for future research to evaluate the nutritional intake of newborns through easily collectible urine samples.

## 5. Conclusions

In this study, targeted and untargeted metabolomics was used to analyze urine samples from breastfed and formula-fed infants. Multivariate statistical analysis methods, including PCA and PLS-DA, confirmed some differentiation between the two groups, with PLS-DA demonstrating a more distinct separation.

To identify potential biomarkers, targeted metabolomics was conducted, analyzing 395 metabolites. Through multivariate analyses, 167 biomarkers were selected, including organic acids, sugars, and amino acids, which played a crucial role in distinguishing between the two feeding groups. Pathway analysis revealed the significant influence of metabolic processes such as amino acid metabolism. Among these biomarkers, 2-hydroxyhippuric acid was significantly increased, indicating potential polyphenol intake from human milk. Significant differences were also observed between the concentrations of urinary amino acids, suggesting that further research could yield a more profound interpretation of these findings.

These findings highlight the intricate metabolic differences between breastfed and formula-fed infants and the potential impact of feeding type on neonatal metabolism, emphasizing the importance of using targeted metabolomics in biomarker discovery and pathway analysis.

The findings from this study suggest that urine metabolomics can be instrumental in assessing differences and qualities in the dietary intake. Urine, being easily collectible and non-invasive compared to other invasive samples such as blood and lymph fluid, offers advantages, making urine metabolite research applicable in various fields. Obtaining samples from a sufficient number of participants in future studies would likely yield more robust data, enhancing the potential for research advancement.

## Figures and Tables

**Figure 1 metabolites-14-00128-f001:**
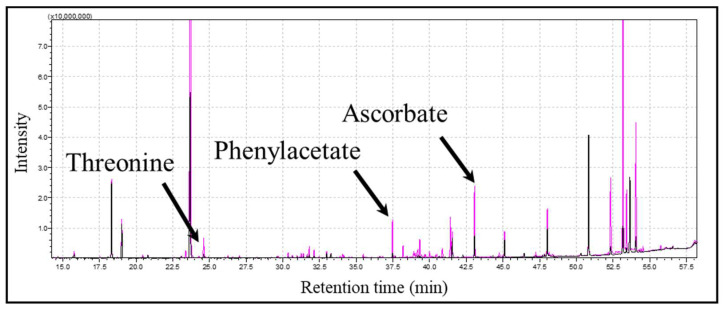
Representative chromatogram from GC-MS (pink: human-milk-fed group, black: formula-milk-fed group).

**Figure 2 metabolites-14-00128-f002:**
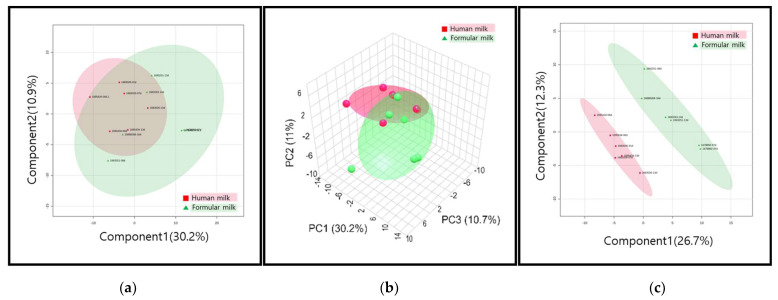
Multivariate statistical analysis of untargeted metabolomics: (**a**) PCA score plot (2D); (**b**) PCA score plot (3D); (**c**) PLS-DA plot.

**Figure 3 metabolites-14-00128-f003:**
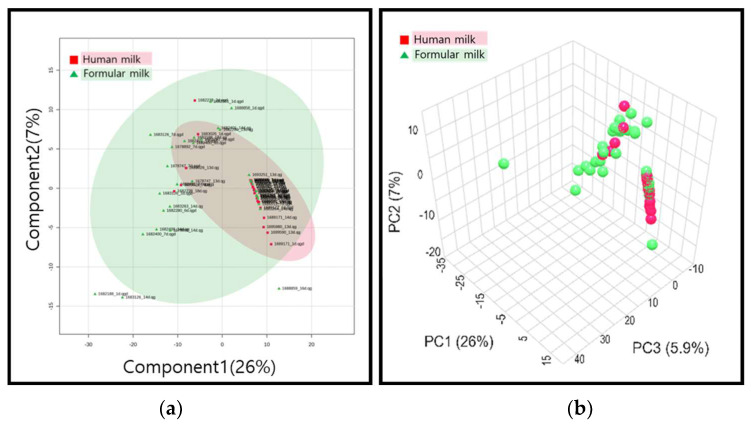
Multivariate statistical analysis of targeted metabolomics: (**a**) PCA score plots (2D); (**b**) PCA score plot (3D); (**c**) PLS-DA plot; (**d**) OPLS-DA plots.

**Figure 4 metabolites-14-00128-f004:**
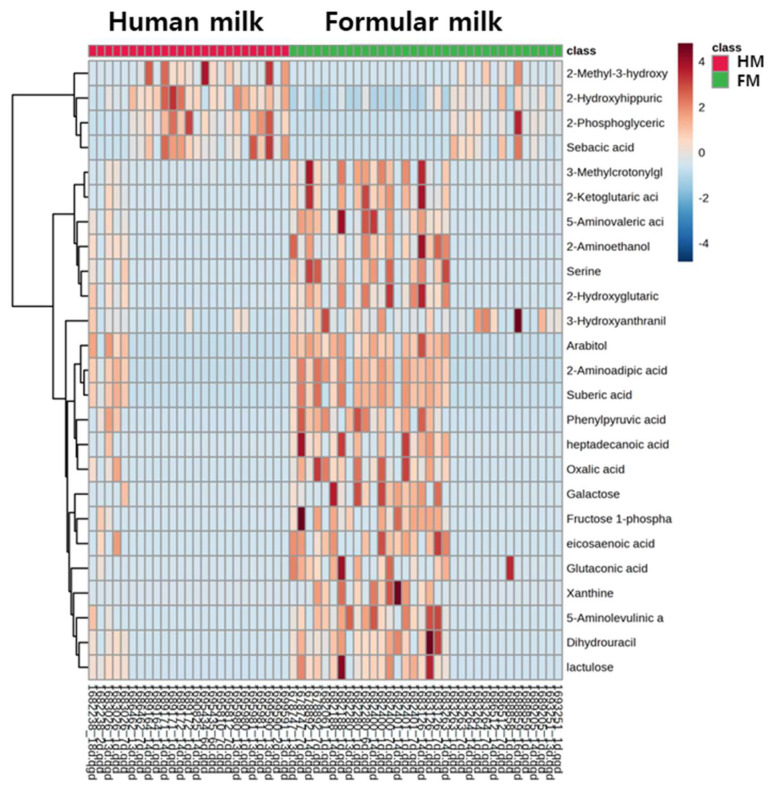
Hierarchical clustering heatmap of targeted metabolomics. The top 25 metabolites with high VIP values.

**Figure 5 metabolites-14-00128-f005:**
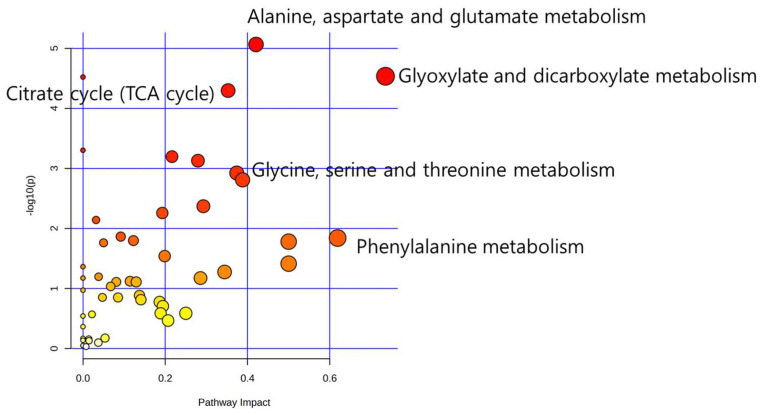
Metabolic pathway analysis. As the distance from the origin increases, the color shifts towards red, indicating a greater influence on metabolic pathway.

## Data Availability

All study data are provided in the manuscript and Appendix A. Detailed methods and additional data are available upon request from the corresponding author.

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
