# Peer review of "Urinary Metabolomic Differentiation of Infants Fed on Human Breastmilk and Formulated Milk"

_metabolites, 2024, doi:10.3390/metabo14020128_

Round 1

Reviewer 1 Report

Comments and Suggestions for Authors

The topic is important and can contribute to the knowledge of metabolism at a crucial time in life.
And It is important because of its technical approach from targeted and untargeted metabolomics.
The references are suitable.
The tables and figures are fine.

some minor comments:
What are the main differences between the biomarkers identified by metabolomics in a group of breastfed and formula-fed neonates?
The methodology needs to be clarified because it is not consistent with the results

The authors show serious contradictions when referring to the methodology vs. when presenting the results. In line 83 they say: "Untargeted metabolomics employed a TQ8050 GC-MS/MS system equipped " and in line 90 they quote: " For targeted metabolomics, analysis was conducted on a Shimadzu GC-MS-TQ8040". But in the presentation of results, lines they say 114 " Untargeted metabolomics using GC-MS " and in line 170 "In this study, targeted metabolomics was conducted using GC-MS/MS in MRM mode " presenting the approximation in the opposite way.

I believe that these contradictions need to be corrected in order to be able to carry out a good review of the findings.

Author Response

Comments 1: The topic is important and can contribute to the knowledge of metabolism at a crucial time in life.

And it is important because of its technical approach from targeted and untargeted metabolomics.

The references are suitable.

The tables and figures are fine.

some minor comments:

What are the main differences between the biomarkers identified by metabolomics in a group of breastfed and formula-fed neonates?

The methodology needs to be clarified because it is not consistent with the results.

The authors show serious contradictions when referring to the methodology vs. when presenting the results. In line 83 they say: "Untargeted metabolomics employed a TQ8050 GC-MS/MS system equipped " and in line 90 they quote: " For targeted metabolomics, analysis was conducted on a Shimadzu GC-MS-TQ8040". But in the presentation of results, lines they say 114 " Untargeted metabolomics using GC-MS " and in line 170 "In this study, targeted metabolomics was conducted using GC-MS/MS in MRM mode " presenting the approximation in the opposite way.

I believe that these contradictions need to be corrected in order to be able to carry out a good review of the findings.

Comments 1: What are the main differences between the biomarkers identified by metabolomics in a group of breastfed and formula-fed neonates?

Response 1: Thank you for pointing this out. The main differences of biomarkers between each group could be summarized with 25 main biomarkers in Figure 2; the increase of 4 metabolites and decrease of 21 metabolites. To emphasize the detailed differences, we add specific trends of biomarkers at line 201-204.

“The heatmap of 25 high VIP metabolites in Figure 2, revealed that limited numbers of metabolites such as 2-hydroxyhippuric acid, and 2-phosphoglyceric acid were increased in breast feeding group, while the levels of 21 other metabolites were down-regulated in the same group.”     

Comments 2: The methodology needs to be clarified because it is not consistent with the results. The authors show serious contradictions when referring to the methodology vs. when presenting the results. In line 83 they say: "Untargeted metabolomics employed a TQ8050 GC-MS/MS system equipped " and in line 90 they quote: " For targeted metabolomics, analysis was conducted on a Shimadzu GC-MS-TQ8040". But in the presentation of results, lines they say 114 " Untargeted metabolomics using GC-MS " and in line 170 "In this study, targeted metabolomics was conducted using GC-MS/MS in MRM mode " presenting the approximation in the opposite way.

Response 2: Thank you for your kind comment. We found that we mistype the instrument “TQ8050 GC-MS/MS” at line 83. We used “GCMS-QP2010SE” which is GC-MS, for untargeted metabolomics. We are sorry for this confusion. We corrected the name at line 83.

“Untargeted metabolomics employed a GCMS-QP2010SE system equipped with a fused silica Rxi-5 ms column (30 m, 0.5 μm film thickness, 0.25 mm ID; Restek Corporation, Bellefonte, PA, USA).”

Reviewer 2 Report

Comments and Suggestions for Authors

The communication manuscript entitled "Urinary Metabolomic Differentiation of Infants Fed on Human Breastmilk and Formulated Milk" (metabolites-2756135) aims to explore the distinctions in urinary metabolites between breastfed and formula-fed infants. This is a fascinating study, yet I have several observations that could enhance its clarity and comprehensibility.

Firstly, in the introduction, it is customary to present the objective at the conclusion, following the hypothesis. The objective detailed from lines 58 to 64 should be rephrased cohesively for improved clarity.

In the Materials and Methods section, it's crucial to specify the study's design, ethics committee approval, and the rationale behind the chosen sample size. Furthermore, the inclusion criteria for the infants and mothers involved in the study need clarification. Given the study's aims, participants in both the breastfed and formula-fed groups must be comparable. This homogeneity ensures that any observed differences can be confidently attributed to the feeding type.

Regarding the results, the graphs provided would benefit from more detailed explanations. Also, it's important to inquire whether the quality of the breast milk from participating mothers was evaluated.

In the discussion section, it is essential to address the study's limitations and outline future research directions. Remember, the discussion is a pivotal part of the paper.

The conclusions represent the contribution of the work to knowledge, rather than merely summarising the tasks completed.

Author Response

Comments: The communication manuscript entitled "Urinary Metabolomic Differentiation of Infants Fed on Human Breastmilk and Formulated Milk" (metabolites-2756135) aims to explore the distinctions in urinary metabolites between breastfed and formula-fed infants. This is a fascinating study, yet I have several observations that could enhance its clarity and comprehensibility.

Firstly, in the introduction, it is customary to present the objective at the conclusion, following the hypothesis. The objective detailed from lines 58 to 64 should be rephrased cohesively for improved clarity.

In the Materials and Methods section, it's crucial to specify the study's design, ethics committee approval, and the rationale behind the chosen sample size. Furthermore, the inclusion criteria for the infants and mothers involved in the study need clarification. Given the study's aims, participants in both the breastfed and formula-fed groups must be comparable. This homogeneity ensures that any observed differences can be confidently attributed to the feeding type.

Regarding the results, the graphs provided would benefit from more detailed explanations. Also, it's important to inquire whether the quality of the breast milk from participating mothers was evaluated.

In the discussion section, it is essential to address the study's limitations and outline future research directions. Remember, the discussion is a pivotal part of the paper.

The conclusions represent the contribution of the work to knowledge, rather than merely summarizing the tasks completed.

Comments 1: Firstly, in the introduction, it is customary to present the objective at the conclusion, following the hypothesis. The objective detailed from lines 58 to 64 should be rephrased cohesively for improved clarity.

Response 1: Thank you for pointing this out. To clarify object of the study, we add sentence about the importance of this investigation at line 60-62.

“The possible impact of breastfeeding was analyzed by urinary metabolism analysis. This study could provide fundamental data to understand urinary metabolome in neonate related to feeding type.”    

Comments 2: In the Materials and Methods section, it's crucial to specify the study's design, ethics committee approval, and the rationale behind the chosen sample size. Furthermore, the inclusion criteria for the infants and mothers involved in the study need clarification. Given the study's aims, participants in both the breastfed and formula-fed groups must be comparable. This homogeneity ensures that any observed differences can be confidently attributed to the feeding type.

Response 2: Thank you for your kind comment. In this study, we approved the research by Institutional Review Board of Soonchunhyang university hospital and we described the detail at line 275-277 (protocol code SCHCA 2020-08-034, 2020.09.01). When conducting the experiment, the sample size was limited by the consent of the participating mothers at the time of sample collection, resulting in the selection of 15 infants each in the breast milk and formula groups. While acknowledging the limitation of a small sample size in this study, efforts were made to maintain homogeneity within each experimental group by ensuring similar distribution of factors such as gender, twin status, and birth timing. Maternal participants and their infants involved in the experiment were selected based on physical indicators such as BMI, head circumference, and blood tests, with only healthy individuals being included in the study. To avoid confusion, we added more detailed description at line 72-76.

“Gender, twin status, and birth timing were distributed similarly in each experimental group to ensure homogeneity. Maternal participants and infants involved in the experiment underwent BMI measurements and blood tests to confirm the absence of specific diseases, and only healthy mothers and infants participated in the study.”

Comments 3: Regarding the results, the graphs provided would benefit from more detailed explanations. Also, it's important to inquire whether the quality of the breast milk from participating mothers was evaluated.

Response 3: Thank you for your accurate comments. In this study, the analytes from each sample were more than 300 metabolites. The selected biomarkers were 25 metabolites. Because of the numerous numbers of variables, we choose heatmap to represent trends of the metabolites rather than graphs. Also, the mothers who participated in this study, are healthy and had average health condition. Because the mothers are health condition was appropriate enough to feed their neonates, the quality of breastmilk could be considered adequate nutritional source to baby. To confirm this, we add mother’s health condition at supplemental data.

Comments 4: In the discussion section, it is essential to address the study's limitations and outline future research directions. Remember, the discussion is a pivotal part of the paper.

Response 4: Thank you for pointing this out. We added limitation and future research direction at line 250-256.

“The study faced limitations in its sample size, as specimens from only 30 newborns were utilized, making generalization of the results challenging. Despite this limitation, statistically significant differences were observed in the urinary metabolites of infants who were breastfed compared to those who were formula-fed, with several biomarkers, in-cluding hippuric acid, identified as contributors to these differences. The study suggests the potential for future research to evaluate the nutritional intake of newborns through easily collectible urine samples.”

Comments 5: The conclusions represent the contribution of the work to knowledge, rather than merely summarizing the tasks completed.

Response 5: Thank you for your kind comment. We added importance and impacts of the study in conclusion part, at line 276-281.

“The findings from this study suggest that urine metabolomics can be instrumental in assessing differences and qualities in the dietary intake. Urine, being easily collectible and non-invasive compared to other invasive samples such as blood and lymph fluid, offers advantages, making urine metabolite research applicable in various fields. Obtaining samples from a sufficient number of participants in future studies would likely yield more robust data, enhancing the potential for research advancement.”

Reviewer 3 Report

Comments and Suggestions for Authors

I have no critical comments.

Author Response

Comments: I have no critical comments.

Round 2

Reviewer 1 Report

Comments and Suggestions for Authors

Although the authors make some clarifications regarding the methodology used and the results obtained, in the general reading of the manuscript some points are perceived that seem to intertwine the idea of targeted and untargeted methodology.

It is noteworthy that the discussion does not address at any time the targeted approach, which, according to the methodology, is the one that, according to the same authors, allows a higher quality in the results. (Line 184:"This highlights the potential of targeted metabolomics to reveal distinct trends in metabolites that were not identified through untargeted metabolomics.")

I believe that the authors should make a general review of the entire manuscript, correct the segments that lend themselves to confusion between the two methodologies and write a conclusion according to the results.

Author Response

Response: Thank you for pointing this out. In this study, untargeted metabolomics provided the advantage of quickly identifying differences between two groups without the need for peak identification. Following the detection of these differences, targeted metabolomics was utilized to derive biomarkers for the specific metabolites that influenced the distinction between the groups. As you suggested, to avoid confusion between the two methodologies, we have revised the statement to align with the workflow accurately. Additionally, we have included interpretations appropriate for each analytical method. (line 186-195)

“In this study, we observed distinct patterns differentiating the breast milk and formula groups through untargeted metabolomics analysis. To identify the metabolites that signif-icantly contributed to this differentiation, we employed targeted metabolomics to pinpoint biomarkers. The results from targeted metabolomics were then emphasized in a statistical analysis using OPLS-DA plots (Figure 3d), which clearly demonstrated a pronounced separation between the two groups.

In summary, untargeted metabolomics is useful for quickly comparing patterns and identifying differences between two groups without specific peak identification. Once dif-ferences are established, targeted metabolomics can be utilized to identify biomarkers that influence the separation between the groups.”     

Reviewer 2 Report

Comments and Suggestions for Authors

I have thoroughly reviewed the updated version of the article titled "Urinary Metabolomic Differentiation of Infants Fed on Human Breastmilk and Formulated Milk" (metabolites-2756135), considering the authors' reply to previous recommendations. I observe that the authors have successfully integrated these recommendations. Nevertheless, I would suggest the inclusion of the ethical committee's approval mentioned in their responses in the manuscript, as it lends greater transparency to the study. Additionally, it is crucial to mention that the sample relies on the voluntary participation of mothers, which is pertinent for the interpretation of the results, it's not a probabilistic sample size.

In summary, the article now presents more lucidly a subject of significant relevance: the metabolomic differences in early infant nutrition.

Author Response

Response 1: Thank you for pointing this out. As you suggested, we have added to the manuscript that the donation of samples was voluntary and had received approval from the ethics committee. (line 76-79) In addition, to enhance the clarity of the research findings, we have strengthened the interpretation of the figures and the conclusion. (line 186-195)

“All the mothers who donated urine for this study did so voluntarily and with consent, and the study was approved by the Research Ethics Committee (Institutional Review Board of Soonchunhyang university hospital, protocol code SCHCA 2020-08-034, 2020.09.01).”    

“In this study, we observed distinct patterns differentiating the breast milk and formula groups through untargeted metabolomics analysis. To identify the metabolites that signif-icantly contributed to this differentiation, we employed targeted metabolomics to pinpoint biomarkers. The results from targeted metabolomics were then emphasized in a statistical analysis using OPLS-DA plots (Figure 3d), which clearly demonstrated a pronounced separation between the two groups.

In summary, untargeted metabolomics is useful for quickly comparing patterns and identifying differences between two groups without specific peak identification. Once dif-ferences are established, targeted metabolomics can be utilized to identify biomarkers that influence the separation between the groups.”
